# General Debiasing for Graph-based Collaborative Filtering via Adversarial Graph Dropout

## ABSTRACT

Graph neural networks (GNNs) have shown impressive performance in recommender systems, particularly in collaborative filtering (CF). The key lies in aggregating neighborhood information on a user-item interaction graph to enhance user/item representations. However, we have discovered that this aggregation mechanism comes with a drawback – it amplifies biases present in the interaction graph. For instance, a user's interactions with items can be driven by both unbiased true interest and various biased factors like item popularity or exposure. But the current aggregation approach combines all information, both biased and unbiased, leading to biased representation learning. Consequently, graph-based recommenders can learn distorted views of users/items, hindering the modeling of their true preferences and generalization.

To address this issue, we introduce a novel framework called Adversarial Graph Dropout (AdvDrop). It differentiates between unbiased and biased interactions, enabling unbiased representation learning. For each user/item, AdvDrop employs adversarial learning to split the neighborhood into two views: one with bias-mitigated interactions and the other with bias-aware interactions. After view-specific aggregation, AdvDrop ensures that the bias-mitigated and bias-aware representations remain invariant, shielding them from the influence of bias. We validate AdvDrop's effectiveness on six public datasets that cover both general and specific biases, demonstrating significant improvements. Furthermore, our method exhibits meaningful separation of subgraphs and achieves unbiased representations for graph-based CF models, as revealed by in-depth analysis. Our code is publicly available at https://anonymous.4open.science/r/INV-LGN-C23D/.

## CCS CONCEPTS

• **Information systems** → **Recommender systems**.

## KEYWORDS

Collaborative Filtering, Popularity Distribution Shift, Debiasing

## 1 INTRODUCTION

Collaborative Filtering (CF) [39] plays a vital role in recommender systems. It basically hypothesizes that behaviorally similar users would share preferences for items [38, 39]. On this hypothesis, graph-based CF has emerged as a dominant line [13, 25, 26, 34, 48, 50, 56], which often recasts user-item interactions as a bipartite graph and exploits it to learn the collaborative signals among users. Hence, employing graph neural networks (GNNs) [19] on the interaction graph becomes a natural and prevalent solution. At the core is hiring multiple graph convolutional layers to iteratively aggregate information among multi-hop neighbors, instantiate the CF signals as the high-order connectivities, and gather them into user and item representations.

Despite the impressive performance of graph-based CF models, we contend that the interaction graph usually contains biased user-item interactions, such as noise-injected observations [12] and missing-not-at-random issue [40]. Worse still, such data biases could be amplified through the GNN mechanism, especially when applying multiple graph convolutional layers [2, 69]. To showcase such a bias amplification, consider a real example in Figures 1 and 2, where LightGCN [26] with zero, two, and four graph convolutional layers (*i.e.,* MF [39], LightGCN-2, and LightGCN-4) is selected as the representative graph-based CF model trained on the Coat dataset [41]. By examining specific factors (*e.g.,* item popularity, user gender) and quantifying prediction disparities across factor groups, we can delineate prediction biases [7] and find:

- Popular items and active users, as highly centralized nodes in the interaction graph, exert a dominant influence over information propagation. For example, we divide item representations into head, middle, and tail groups, based on popularity. With the layer depth increase from zero to four (*cf.* Figures 1e-1g), head items cluster closer at the origin while tail items keep spreading outwardly, indicating that the GNN mechanism encourages the domination of popular items.
- Conformity and sensitive attributes of users, even not used during training, exhibit over-aggregation through multi-hop neighbors. This results in undesirable clustering of user representations — a clear indication of conformity bias [67] and fairness issues [7, 36]. Take user gender as an example of a sensitive attribute. With the increase of layer depth from zero to four (*cf.* Figures 1a-1c), user representations are more compactly clustered *w.r.t.* two gender groups.

Conclusively, data biases risk amplification via the GNN mechanism [65], potentially leading a GNN backbone to generate representations more biased than MF counterparts. This not only compromises out-of-distribution generalization in wild environments [3, 29], but may also increase the risks of leaking private attributes.

In response to such negative influences, recent debiasing strategies [7, 11, 52, 53, 55, 67] have emerged. However, we argue that there are several limitations: (1) Data Bias Perspective: Typically, only one specific factor is considered to be mitigated, such as item popularity [15, 23, 55], user conformity [67], and sensitive attributes [7]. However, interaction data is often riddled with various biases, whether predefined or arising from latent confounders. (2) Bias Amplification Perspective. The debiased GNN mechanism is tailor-made for a particular bias. For instance, tailored specifically for popularity bias, they randomly sample interaction subgraphs [56] and alter the information propagation scheme [68], but might fail in other biases. Here we argue that, for graph-based CF models, effective debiasing should comprehensively address both biases present in interaction data and those amplified by the GNN mechanism. While conceptually compelling, such general debiasing remains


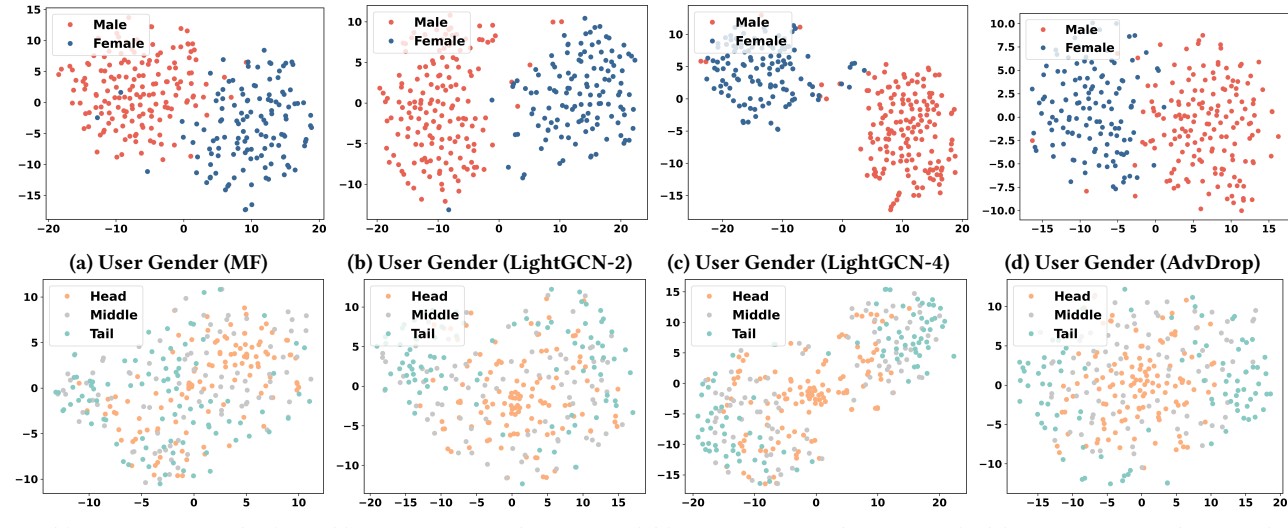

**Figure 1: T-SNE [44] visualizations of user and item representations learned by MF [39], LightGCN [26], and our proposed AdvDrop. Note that MF, LightGCN-2, and LightGCN-4 are specialized with zero, two, and four graph convolutional layers, respectively. Subfigures 1a-1d show the representation distribution *w.r.t.* two groups of user gender (*i.e.,* female, male), while Subfigures 1e-1h depict the representation distribution *w.r.t.* three groups of item popularity (*i.e.,* head, middle, tail).**

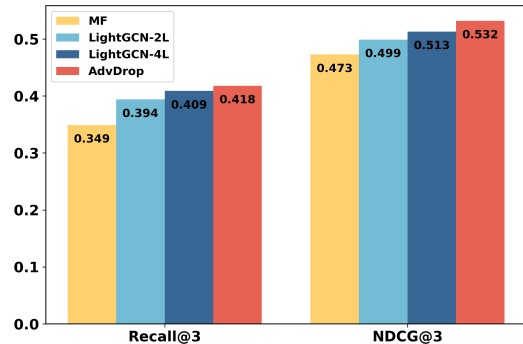

**Figure 2: Recommendation Performance**

largely unexplored in literature [47] — a gap our work seeks to bridge.

This motivates us to develop a general debiasing strategy for graph-based CF models, which can autonomously identify and eliminate biases during information propagation. We anchor our approach on the principles of invariant learning [3, 29, 64], which encourages the representation learning to best support the prediction, while remaining invariant to varying factors. Hence, we propose Adversarial Graph Dropout (**AdvDrop**). Specifically, by learning to employ the edge dropout on the interaction graph, it adversarially identifies bias-mitigated and bias-aware subgraphs, and makes the representations derived from one subgraph invariant to the counterparts from the other subgraph. By framing the bias-aware subgraph generation as the bias identification stage and the invariant learning as the debiasing stage, our AdvDrop is distilled into an iterative optimization process under a min-max framework:

- In the debiased representation learning stage, AdvDrop refines user/item representations by concurrently optimizing the recommendation objective and **minimizing** the discrepancy between

the representations learned from sampled bias-mitigated and bias-aware subgraphs.
- In the bias identification stage, AdvDrop determines the bias distributions of interaction in an adversarial manner by **maximizing** discrepancy between user/item representations that come from sampled subgraphs.

As a result, AdvDrop is able to simultaneously discover various biases during information propagation, and achieve debiased representation learning, thus acting as a simple yet effective debiasing plugin for graph-based CF models. Empirical validations, especially in datasets characterized by the general bias and multiple specific biases, show that our AdvDrop consistently surpasses leading debiasing baselines (*e.g.,* CVIB [52], InvPref [53], sDRO [55]). The improvements in recommendation accuracy (*cf.* Figure 1) coupled with unbiased distributions *w.r.t.* bias factors (*cf.* Figures 1d and 1h) clearly demonstrate the effectiveness of AdvDrop.

Our main contributions are summarized as:

- We uncover the problem of model inherent bias for graph-based CF models, which is important and widespread but far from being well-studied.
- We propose a new debiasing framework in an adversarial manner, named as AdvDrop, which forces the representations learned from different subgraphs generated by bias-aware dropout to be consistent with each other.
- Through extensive experiments including both general and specific bias scenarios as well as visualization of learned representations and bias distributions, we validate the universal effectiveness of AdvDrop.

## 2 PRELIMINARY

We start by presenting an overview of CF and the general framework of graph-based CF.

## 2.1 Collaborative Filtering (CF)

Collaborative Filtering (CF) [39] is fundamental in recommender systems, which basically assumes that users with similar behavioral patterns are likely to exhibit shared preferences for items. Here we focus on CF with implicit feedback (*e.g.*, clicks, views, purchases) [38, 39]. Let $\mathcal{U}$ and $\mathcal{I}$ be the sets of users and items, respectively. Each user $u \in \mathcal{U}$ has interacted with a set of items denoted by $\mathcal{I}_u^+$, while $\mathcal{I}_u^- = \mathcal{I} - \mathcal{I}_u^+$ collects her/his non-interacted items. Such user-item interactions can be summarized as the matrix $\mathbf{Y}$, where $y_{ui} = 1$ indicates that user $u$ has adopted item $i$, and $y_{ui} = 0$ indicates no interaction.

Following prior studies [25, 26], we can construct a bi-partite graph $\mathcal{G} = (\mathcal{V}, \mathcal{E})$, by combining all users and items into the node set $\mathcal{V} = \mathcal{U} \cup \mathcal{I}$, and treating their observed interactions as the edges $\mathcal{E}$. The adjacency matrix $\mathbf{A}$ of $\mathcal{G}$ can be derived as follows:

$$\mathbf{A} = \begin{bmatrix} \mathbf{0} & \mathbf{Y} \\ \mathbf{Y}^{\mathsf{T}} & \mathbf{0} \end{bmatrix}. \tag{1}$$

The primary goal of recommendation is to identify a list of items that potentially align with the preferences of each user $u$. This task can be viewed as a link prediction problem within the bipartite graph $\mathcal{G}$.

## 2.2 Graph-based CF Framework

Graph-based CF aims to leverage the bipartite interaction graph, capturing the collaborative signals between users and thereby predicting user preferences towards items. To this end, employing graph neural networks (GNNs) [19] on the graph becomes an intuitive and widely accepted strategy. At the core is, for each ego user/item node, employing multiple graph convolutional layers to learn the representation. These layers function to iteratively aggregate information among multi-hop neighbors, instantiate the CF signals as the higher-order connectivities, and subsequently gather them into the user/item representations. Formally, the information aggregation can be summarized as:

$$\mathbf{Z}^{(l)} = \text{AGG}(\mathbf{Z}^{(l-1)}, \mathbf{A}), \tag{2}$$

where $\mathbf{Z}^{(l)}$ represents the node representations after $l$ graph convolutional layer, and $\mathbf{Z}^{(0)}$ is initialized as the embeddings of users and items [26, 48]. The aggregation function $\text{AGG}(\cdot)$ is to integrate the useful information from a node's neighbors to refine its representation. Taking a user $u$ as an example, the aggregation function can be expressed as:

$$\mathbf{z}_u^{(l)} = f_{\text{combine}}(\mathbf{z}_u^{(l-1)}, f_{\text{aggregate}}(\{\mathbf{z}_i^{(l-1)} | \mathbf{A}_{ui} = 1\})), \tag{3}$$

where $u$'s representation at the $l$-th layer, $\mathbf{z}_u^{(l)}$, is obtained by first aggregating her/his neighbors' representations from the $(l-1)$-th layer via $f_{\text{aggregate}}(\cdot)$, and then combining with her/his own representation $\mathbf{z}_u^{(l-1)}$ via $f_{\text{combine}}(\cdot)$. Finally, we employ a readout function on the user representations derived from different layers to get the final representation:

$$\mathbf{z}_u = f_{\text{readout}}(\{\mathbf{z}_u^{(l)} | l = [0, \cdots, L]\}). \tag{4}$$

Analogously, we can get an item $i$'s final representation, $\mathbf{z}_i$. For brevity, we summarize the procedure of applying GNN on graph $\mathcal{G}$ to obtain aggregated representations of users/items as a single equation:

$$\mathbf{Z}_U, \mathbf{Z}_I = \text{GNN}(\mathcal{G}|\Theta), \tag{5}$$

where $\mathbf{Z}_U$ and $\mathbf{Z}_I$ collect representations of all users and items, respectively; $\Theta$ represents all trainable parameters.

Having obtained the final representations of a user $u$ and an item $i$, we can build a similarity function upon them to predict how likely $u$ will interact with $i$:

$$\hat{y}_{ui} = s(\mathbf{z}_u, \mathbf{z}_i), \tag{6}$$

where the similarity function $s(\cdot)$ can be set as inner product, cosine similarity, and neural networks; the predictive score $\hat{y}_{ui}$ indicates the preference of user $u$ towards item $i$.

To learn the model parameters, we utilize the observed interactions as the supervision signal. Then, we encourage the predictive score $\hat{y}_{ui}$ to either align with the ground-truth value $y_{ui}$ for point-wise learning [42] or preserve the preference order of $\{y_{ui} | i \in \mathcal{N}_u\}$ for pair-wise [39] and list-wise learning [9]. In this work, we adopt the Bayesian Personalized Ranking (BPR) loss [39] as our training objective, which is a widely-used choice for pair-wise learning:

$$\mathcal{L}_{\text{BPR}} = \sum_{(u,i,j) \in O} -\log \sigma(\hat{y}_{ui} - \hat{y}_{uj}), \tag{7}$$

where $O = \{(u, i, j) | u \in \mathcal{U}, i \in \mathcal{I}_u^+, j \in \mathcal{I}_u^-\}$ is the training data. This objective enforces that the prediction of an observed interaction should receive a higher score than its unobserved counterparts. This serves as our primary supervision signal during training.

## 3 METHODOLOGY

Here we present our Adversarial Graph Dropout (AdvDrop) framework, as Figure 3 illustrates. It aims to combat the intrinsic biases emerging during graph-based CF. See the comprehensive literature review on debiasing in Appendix A.1. Specifically, our AdvDrop comprises two training stages, mapping to a min-max optimization:

- **Debiased Representation Learning**. To mitigate biases during representation learning, we employ a bias measurement function $P_B$ to quantify the bias of each interaction. Leveraging $P_B$, we construct two bias-related views of nodes and aim for embedding-level invariance to varying biases. Specifically, we create bias-aware and bias-mitigated subgraphs by performing random edge dropout according to $P_B$ and $1 - P_B$, respectively. Neighbor aggregation is then separately performed on these subgraph views. On the view-specific representations, a contrastive learning loss is minimized between them to ensure representation-level invariance, alongside the primary recommendation objective (*e.g.*, Equation (7)).

- **Bias Identification**. To obtain the bias measurement function $P_B$, the AdvDrop framework proposes an adversarial learning approach. Specifically, we learn $P_B$ by adversarially maximizing the divergence between the bias-aware and bias-mitigated node representations. This enables us to identify and quantify the bias in the recommendation system.

By alternating between these stages, AdvDrop not only identifies biases but also mitigates them iteratively. In the following sections, we will provide further details on each of these stages.

Figure 3: The overall framework of AdvDrop.

## 3.1 Debiased Representation Learning

To quantify bias within the interaction graph $\mathcal{G}$ and amplified through the GNN mechanism, AdvDrop incorporates a learnable bias measurement function $P_B$. The level of bias associated with an interaction between a user $u$ and an item $i$ is represented as follows:

$$b_{ui} = P_B(u, i). \tag{8}$$

The bias measurement function $P_B$ outputs values within $[0, 1]$, reflecting the extent to which an interaction is affected by bias factors. A score of 1 indicates that the interaction is entirely biased, whereas a score of 0 suggests that the interaction is solely based on the user's genuine preference. We will elaborate on how to obtain this bias measurement function $P_B$ in the Bias Identification stage (*cf.* Section 3.2).

Using the bias measurement function $P_B$, we derive two distinct views of the interaction graph: bias-aware $\mathcal{G}^+$ and bias-mitigated $\mathcal{G}^-$. Specifically, these views are subgraphs of the original interaction graph $\mathcal{G}$, obtained by performing edge dropout according to $P_B$ and $1 - P_B$. Their respective adjacency matrices are defined as:

$$\begin{aligned} \mathbf{A}^+ &= \mathbf{A} \odot \mathbf{M}^+, \\ \mathbf{A}^- &= \mathbf{A} \odot \mathbf{M}^-, \end{aligned} \tag{9}$$

where $\mathbf{M}^+$ and $\mathbf{M}^-$ are the view-specific masks; the element-wise product $\odot$ is applied to perform the edge dropout based on these masks, resulting in the bias-aware and bias-mitigated subgraphs. Specifically, $\mathbf{M}^+$ and $\mathbf{M}^-$ are constructed such that each edge $(u, i)$'s masks $m_{ui}^+$ and $m_{ui}^-$ follow a Bernoulli distribution with parameters $P_B(u, i)$ and $1 - P_B(u, i)$, respectively:

$$\begin{aligned} \mathbf{M}^+ &\sim P_B(\mathcal{G}^+) = \prod_{\{(u,i)|A_{ui}=1\}} \mathrm{Bern}(m_{ui}^+; P_B(u, i)), \\ \mathbf{M}^- &\sim P_B(\mathcal{G}^-) = \prod_{\{(u,i)|A_{ui}=1\}} \mathrm{Bern}(m_{ui}^-; 1 - P_B(u, i)). \end{aligned} \tag{10}$$

At the beginning of each training epoch, we sample the masks using Equation (10) and then leverage them to create the bias-aware and bias-mitigated views, $\mathcal{G}^+$ and $\mathcal{G}^-$, based on Equation (9). During training, the GNN encoder within the graph-based CF models, characterized by parameters $\Theta_E$, is separately applied to

these two views like Equation (5). As a result, we obtain the view-centric representations for both users and items from each view:

$$\begin{aligned} \mathbf{Z}_U^+, \mathbf{Z}_I^+ &= \mathrm{GNN}(\mathcal{G}^+ | \Theta_E), \\ \mathbf{Z}_U^-, \mathbf{Z}_I^- &= \mathrm{GNN}(\mathcal{G}^- | \Theta_E). \end{aligned} \tag{11}$$

Following SimCLR [14], we use the InfoNCE loss as an auxiliary objective to enforce the representation-level invariance [3, 29, 53, 64] — that is, encouraging the consistency in representations of the same node across both views, while distinguishing between representations of different nodes:

$$\begin{aligned} \mathcal{L}_U^{\mathrm{inv}} &= \sum_{u \in \mathcal{U}} -\log \frac{\exp(s(\mathbf{z}_u^+, \mathbf{z}_u^- / \tau))}{\sum_{u' \in \mathcal{U}} \exp(s(\mathbf{z}_u^+, \mathbf{z}_{u'}^-))/\tau)}, \\ \mathcal{L}_I^{\mathrm{inv}} &= \sum_{i \in \mathcal{I}} -\log \frac{\exp(s(\mathbf{z}_i^+, \mathbf{z}_i^- / \tau))}{\sum_{i' \in \mathcal{I}} \exp(s(\mathbf{z}_i^+, \mathbf{z}_{i'}^-))/\tau)}, \end{aligned} \tag{12}$$

where $s(\cdot)$ is the similarity function. For a user/item node, promoting consistency between its bias-aware and bias-mitigated representations allows the model to capture the invariant signal, regardless of the variations caused by bias. This aids in distilling essential user preferences while mitigating the impact of bias on the learned representations. The final contrastive loss is obtained by combining the two terms:

$$\mathcal{L}^{\mathrm{inv}} = \mathcal{L}_U^{\mathrm{inv}} + \mathcal{L}_I^{\mathrm{inv}}, \; \mathbf{M}^+ \sim P_B(\mathcal{G}^+), \; \mathbf{M}^- \sim P_B(\mathcal{G}^-). \tag{13}$$

We then combine this contrastive loss with the recommendation loss, utilizing BPR on the bias-aware and bias-mitigated graphs as Equation (7): $\mathcal{L}^{\mathrm{rec}} = \mathcal{L}_{\mathrm{BPR}}^+ + \mathcal{L}_{\mathrm{BPR}}^-$. Consequently, the final objective of the Debiased Representation Learning stage is formalized as:

$$\min_{\Theta_E} \mathcal{L}^{\mathrm{rec}} + \lambda \mathcal{L}^{\mathrm{inv}}, \tag{14}$$

where $\lambda$ is a hyperparameter to control the strength of representation-level invariance, and $\Theta_E$ collects the model parameters of GNNs.

## 3.2 Bias Identification

To identify the underlying bias of each interaction, AdvDrop adopts an adversarial approach in learning the bias measurement function

---

**Algorithm 1** AdvDrop Algorithm for General Debias

---

**Input:** Training dataset $\mathcal{D}$ consist of $(u, v)$ pairs, the number of stage 1 epochs $K_{stage1}$, and number of stage 2 epochs $K_{stage2}$.
**Output:** Debiased user representations $\mathbf{Z}_U$ and debiased item representations $\mathbf{Z}_I$.
**Initialize:** initialize $\Theta_E$ and $\Theta_B$.
**while** not converged **do**
  **Stage 1:** Debiased Representation Learning:
  Fix bias measurement function parameters $\Theta_B$.
  **for** $k_1 \le K_{stage1}$ **do**
    Compute $\mathcal{L}^{rec}$ and $\mathcal{L}^{inv}$ by Equations (7) and (13).
    Update $\Theta_E$ by Equation (14).
    $k_1 \leftarrow k_1 + 1$
  **end for**
  **Stage 2:** Bias Identification:
  Fix graph neural networks parameters $\Theta_E$.
  **for** $k_2 \le K_{stage2}$ **do**
    Recompute $P_B$ and resample $\mathcal{G}_-$ and $\mathcal{G}_+$.
    Compute $\nabla_{f_B} \mathcal{L}^{inv}$ by Equation (19) and Corollary 3.2.
    Obtain gradients by back-propagation and update $\Theta_B$.
    $k_2 \leftarrow k_2 + 1$
  **end for**
**end while**
Compute $\mathbf{Z}_U$ and $\mathbf{Z}_I$ by Equation (20).
**return** $\mathbf{Z}_U$ and $\mathbf{Z}_I$.

---

$P_B$. Here, we first present the function as:

$$P_B(u, i|\Theta_B) = \sigma(f_B(\mathbf{Z}|\Theta_B)) = \sigma(\mathbf{W}_B[\mathbf{z}_u^{(0)}||\mathbf{z}_i^{(0)}] + b_B), \quad (15)$$

where $\mathbf{z}_u^{(0)}$ and $\mathbf{z}_i^{(0)}$ represent the base embeddings of user $u$ and item $i$ at layer 0, respectively. The operator $(\cdot||\cdot)$ denotes vector concatenation, and $\Theta_B = (\mathbf{W}_B, b_B)$ represents the parameters of the neural network. Importantly, this design allows $P_B(u, i) \ne P_B(i, u)$, which means that the bias measurement might differ when considering the message passing from user-to-item versus item-to-user. Such distinction accounts for potential differences in biases between users and items during the messaging process.

Building upon the defined bias measurement function, we draw inspiration from the adversarial environment inference [18] and directly maximize the contrastive learning loss (*cf.* Equation (13)):

$$\max_{\Theta_B} \mathcal{L}^{inv}. \quad (16)$$

Intuitively, this maximization prompts the bias measurement function to learn two distinct edge drop distributions, which in turn enlarges the representation discrepancy across the two views. We find that training with this objective yields a meaningful bias measurement function suitable for different training stages, empowering the Debias Representation Learning objective to iteratively mitigate bias. See Section 4 for further interpretation of $P_B$.

## 3.3 Model Optimization for AdvDrop

We integrate the loss functions of both learning stages into a unified objective as:

$$\min_{\Theta_E}[\mathcal{L}^{rec} + \lambda \max_{\Theta_B} \mathcal{L}^{inv}]. \quad (17)$$

During training, we first fix the bias measurement function's parameters $\Theta_B$ and perform optimization on the GNNs' parameters $\Theta_E$ in the Debias Representation Learning stage, then fix $\Theta_E$ and optimize $\Theta_B$ in the Bias Identification stage.

However, direct optimization of $\Theta_B$ presents challenges owing to the discrete nature of sampled $\mathbf{M}^+$ and $\mathbf{M}^-$. To address this, we adopt augment-REINFORCE-merge (ARM), a recently proposed unbiased gradient estimator for stochastic binary optimization [61]. Specifically, the key theorem for ARM is shown as follows:

**Theorem 3.1.** *For a vector of $N$ binary random variables $\mathbf{x} = (x_1, \ldots, x_N)^T$, and any function $f$, the gradient of*

$$\mathcal{E}(\boldsymbol{\phi}) = \mathbb{E}_{\mathbf{x} \sim \prod_{n=1}^N \text{Bern}(x_n; \sigma(\phi_n))}[f(\mathbf{x})]$$

*with respect to $\boldsymbol{\phi} = (\phi_1, \ldots, \phi_N)^T$, the logits of the Bernoulli probability parameters, can be expressed as:*

$$\nabla_{\boldsymbol{\phi}} \mathcal{E}(\boldsymbol{\phi}) = \mathbb{E}_{\boldsymbol{v} \sim \prod_{n=1}^N \text{Uniform}(v_n; 0,1)} \Big[ (f(\mathbb{I}[\boldsymbol{v} > \sigma(-\boldsymbol{\phi})]) $$
$$- f(\mathbb{I}[\boldsymbol{v} < \sigma(\boldsymbol{\phi})]))(\boldsymbol{v} - \frac{1}{2}) \Big],$$

*where $\mathbb{I}[\boldsymbol{v} > \sigma(-\boldsymbol{\phi})] := (\mathbb{I}[v_1 > \sigma(-\phi_1)], \ldots, \mathbb{I}[v_N > \sigma(-\phi_N)])^T$, and $\sigma(\cdot)$ is the sigmoid function.*

This theorem provides an unbiased estimator for gradients of Bernoulli variables, requiring merely an antithetically coupled pair of samples drawn from the uniform distribution. In alignment with Theorem 3.1, we sample two random variables $\boldsymbol{v}_1$ and $\boldsymbol{v}_2$ from uniform distribution $U_{\mathcal{G}} = \prod_{\{(i,j)|A_{ij}=1\}} \text{Uniform}(v_{ij}, 0, 1)$. The contrastive loss, based on $\boldsymbol{v}_1$ and $\boldsymbol{v}_2$, is defined as:

$$\mathcal{L}_{P_B,>}^{inv} := (\mathcal{L}_U^{inv} + \mathcal{L}_I^{inv})\Big|_{\substack{M_+ = \mathbb{I}[\boldsymbol{v}_1 > \sigma(-f_B(\mathbf{Z}|\Theta_B))], \\ M_- = \mathbb{I}[\boldsymbol{v}_2 > \sigma(f_B(\mathbf{Z}|\Theta_B))]}}$$
$$\mathcal{L}_{P_B,<}^{inv} := (\mathcal{L}_U^{inv} + \mathcal{L}_I^{inv})\Big|_{\substack{M_+ = \mathbb{I}[\boldsymbol{v}_1 < \sigma(f_B(\mathbf{Z}|\Theta_B))], \\ M_- = \mathbb{I}[\boldsymbol{v}_2 < \sigma(-f_B(\mathbf{Z}|\Theta_B))]}} . \quad (18)$$

Then an unbiased estimation of gradients can be computed according to the following corollary of Theorem 3.1:

**Corollary 3.2.** *The gradient of contrastive loss $\mathcal{L}^{inv}$ in AdvDrop with respect to logits of the Bernoulli probability parameters $f_B$ can be expressed as:*

$$\nabla_{f_B} \mathcal{L}^{inv} = \mathbb{E}_{\boldsymbol{v}_1, \boldsymbol{v}_2 \sim U_{\mathcal{G}}} \Big[ (\mathcal{L}_{P_B,>}^{inv} - \mathcal{L}_{P_B,<}^{inv})(\boldsymbol{v}_1 - \boldsymbol{v}_2) \Big], \quad (19)$$

*where $U_{\mathcal{G}} = \prod_{\{(i,j)|A_{ij}=1\}} \text{Uniform}(v_{ij}, 0, 1)$, $\mathcal{L}_{P_B,>}^{inv}$ and $\mathcal{L}_{P_B,>}^{inv}$ are defined according to Equation 18.*

The gradients of $\Theta_B$ can thus be obtained by back-propagation after obtaining the gradients *w.r.t.* $f_B$. During inference, we compute the representations $\mathbf{Z}_U$ and $\mathbf{Z}_I$ respectively without graph dropout:

$$\mathbf{Z}_U, \mathbf{Z}_I = \text{GNN}(\mathcal{G}|\Theta_E). \quad (20)$$

The overall algorithm of AdvDrop is summarized in Algorithm 1.

## 4 EXPERIMENTS

To validate the effectiveness of AdvDrop, we conduct extensive experiments targeting the following research queries::

- **RQ1:** How does Advdrop perform compared with other baseline models on general debiasing datasets?
- **RQ2:** Can Advdrop successfully address various specific biases?

**Table 1: General debiasing performance on Coat, Yahoo, and KuaiRec. The top-performing method for each metric is highlighted in bold, with the runner-up underlined. The improvements achieved by AdvDrop are statistically significant ($p$-value $\ll 0.05$).**

| | Coat | | Yahoo | | KuaiRec | |
|---|---|---|---|---|---|---|
| | NDCG@3 | Recall@3 | NDCG@3 | Recall@3 | NDCG@20 | Recall@20 |
| LightGCN | 0.499 | 0.394 | 0.610 | 0.640 | 0.334 | 0.073 |
| IPS-CN | $0.516^{+3.41\%}$ | $0.406^{+3.05\%}$ | $0.598^{-1.97\%}$ | $0.628^{-1.88\%}$ | $0.014^{-95.81\%}$ | $0.002^{-97.26\%}$ |
| DR | $0.506^{+1.40\%}$ | $0.416^{+5.58\%}$ | $0.611^{+0.16\%}$ | $0.637^{-0.47\%}$ | $0.037^{-88.92\%}$ | $0.010^{-86.30\%}$ |
| CVIB | $0.488^{-2.20\%}$ | $0.386^{-2.03\%}$ | $0.597^{-2.13\%}$ | $0.632^{-1.25\%}$ | $0.342^{+2.40\%}$ | $0.079^{+8.22\%}$ |
| InvPref | $0.365^{-26.85\%}$ | $0.293^{-25.63\%}$ | $0.594^{-2.62\%}$ | $0.621^{-2.97\%}$ | - | - |
| AutoDebias | $0.502^{+0.60\%}$ | $0.401^{+1.78\%}$ | $0.601^{-1.48\%}$ | $0.627^{-2.03\%}$ | $0.327^{-2.10\%}$ | $0.072^{-1.37\%}$ |
| **AdvDrop** | $\mathbf{0.532}*^{+6.61\%}$ | $\mathbf{0.418}*^{+6.09\%}$ | $\mathbf{0.617}*^{+1.15\%}$ | $\mathbf{0.643}*^{+0.47\%}$ | $\mathbf{0.362}*^{+8.38\%}$ | $\mathbf{0.089}*^{+21.92\%}$ |

- **RQ3:** Within AdvDrop, what pivotal insights does the adversarial learning framework extract, and how do these influence the learned representations?

## 4.1 Experimental Settings

**Datasets.** We perform experiments on four real-world benchmark datasets, spanning both general and specific bias settings: Yahoo [35], Coat [41], KuaiRec [21], and Yelp2018 [26] for item. See Appendix A.2 for the dataset details and Table 6 for the data statistics.

**Evaluation Metrics.** To evaluate the performance of the model, we use three metrics: NDCG@K [64], Recall@K [64], and Prediction bias [7]. See Appendix A.3 for the metric details.

**Baselines.** We adopt various baselines tailored for mitigating general biases and specific biases. See Appendix A.4 for the details.

- **Mitigating General Bias**: IPS-CN [24], DR [51], CVIB [52], InvPref [53], AutoDeibas [11];
- **Mitigating Specific Biases**: IPS-CN [24] for item popularity, CDAN [16] for item popularity, sDRO [55] for item popularity, and CFC [6] for attribute unfairness.

## 4.2 Performance *w.r.t.* General Bias (RQ1)

**Motivation.** Current recommendation debiasing approaches, targeting general bias, primarily focus on an unbiased test set derived from completely missing-at-random (MAR) user feedback. Yet, in real-world applications, an effective general debiasing algorithm should excel in situations with unidentified distribution shifts in user-item interactions, such as temporal or demographic shifts. In our experiments, we assess the conventional MAR setting in Yahoo & Coat, without any prior knowledge of the test distribution.

**Results.** Table 1 presents the general debiasing performance of AdvDrop in contrast with various baselines. See Table 5 in the appendix for more results. The results yield the following insights::

- **AdvDrop consistently outperforms various baselines for general debiaing across the benchmark datasets.** Specifically, on Coat, Yahoo, and KuaiRec, it achieves relative improvements of 6.61%, 1.15%, and 8.38% in NDCG compared to the LightGCN backbone. These improvements are more significant than those observed with other debiasing baselines. We ascribe the robust performance across diverse bias scenarios to AdvDrop's ability to iteratively identify the interaction bias from the interaction graph, capture the bias amplification within

GNN mechanism, and adversarially mitigate them without prior assumptions of the test distribution.

- **General debiasing baselines exhibit varying performances across MAR and temporal-split settings, which might be critically affected by the model-inherent biases.** Compared to AdvDrop which is tailored to mitigate both general biases and bias amplification inherent in the GNN mechanism, most baselines tend to overlook the biases introduced by the GNN itself. A closer look reveals: IPS-CN excels in the small-scale MAR setting of Coat, but underperforms in other contexts; DR performs well in MAR settings, but fails when applied to temporal distribution shift, mainly due to the misestimations of observation probabilities on unknown distributions; CVIB consistently underperforms the LightGCN backbone on Yahoo. These results underscore the importance of direct interaction bias learning without presupposing test distribution or bias factors. Surprisingly, while InvPref paired with LightGCN struggles to capture the general bias across all datasets, the combination of MF and InvPref effectively addresses this issue [53]. This discrepancy implies the latent biases within the GNN mechanism, further emphasizing the importance of considering both data-centric and GNN-inherent biases.

## 4.3 Performance on Specific Bias (RQ2)

**Motivation.** Intuitively, general debiasing strategies should be able to handle a range of specific biases that exist in recommendation scenarios, performing at par with strategies designed for particular biases. To further validate AdvDrop's capabilities, we conduct experiments focusing on two prevalent bias-related challenges: popularity bias and attribute unfairness. Resolving popularity bias requires the model to perform well when facing popularity-related distribution shifts, while addressing attribute unfairness emphasizes both representation-level and prediction-level parity for sensitive user or item attributes. These two scenarios together demand the OOD generalization, while ensuring unbiased predictions and representations.

*4.3.1 Popularity Bias.* Table 3 presents the results dealing with popularity bias. We can observe that AdvDrop achieves significant improvements over the LightGCN backbone on both ID and OOD test sets, outperforming all compared baselines. With further analysis of AdvDrop in Section 4.4.1, we attribute the ID performance gain primarily stems from the contrastive objective within the Debiased Representation Learning phase, and the OOD performance

**Table 2: Performance of mitigating attribute unfairness on Coat.**

| Evaluation Metrics | Performance Metrics | | | | Fairness Metric (Prediction Bias) | | |
|---|---|---|---|---|---|---|---|
| | NDCG@3 | NDCG@5 | Recall@3 | Recall@5 | user gender | item colour | item gender |
| MF | 0.473 | 0.508 | 0.349 | 0.501 | 0.102 | 0.175 | 0.091 |
| LGN | 0.499 | 0.523 | 0.394 | 0.519 | 0.420 | 0.589 | 0.468 |
| AdvDrop | **0.532** | 0.553 | 0.418 | 0.540 | 0.142 | 0.053 | 0.062 |
| +Embed info | 0.518 | **0.560** | 0.418 | 0.578 | 0.052 | 0.032 | 0.046 |
| +Mask info | 0.512 | 0.554 | **0.425** | **0.581** | 0.045 | 0.024 | 0.039 |
| CFC | 0.485 | 0.513 | 0.395 | 0.517 | **0.012** | **0.011** | **0.012** |

**Table 3: Performance of mitigating popularity bias on Yelp2018.**

| Test Split | Test ID | | Test OOD | |
|---|---|---|---|---|
| Metrics | NDCG@20 | Recall@20 | NDCG@20 | Recall@20 |
| LightGCN | 0.0371 | 0.0527 | 0.0028 | 0.0026 |
| IPS-CN | 0.0337 | 0.0470 | 0.0033 | 0.0030 |
| CDAN | 0.0496 | 0.0703 | 0.0037 | 0.0037 |
| sDRO | 0.0492 | 0.0702 | 0.0035 | 0.0034 |
| AdvDrop | **0.0608** | **0.0817** | **0.0066** | **0.0073** |

**Table 4: Ablation study of AdvDrop on Yelp2018.**

| Test Split | Test ID | | Test OOD | |
|---|---|---|---|---|
| Metrics | NDCG@20 | Recall@20 | NDCG@20 | Recall@20 |
| AdvDrop | **0.0608** | **0.0817** | **0.0066** | **0.0073** |
| w/o $P_B$ | 0.0575 | 0.0781 | 0.0060 | 0.0060 |
| w/o $P_B$ & $\mathcal{L}_{inv}$ | 0.0371 | 0.0528 | 0.0027 | 0.0024 |

gain can be credited to both Debias Representation Learning and Bias Identification stages.

*4.3.2 Attribute Unfairness.* Table 2 delineates the results addressing the attribute unfairness. We evaluate debiasing strategies against the MF and LightGCN backbones by analyzing both the recommendation performance and the fairness metric related to group-wise prediction bias. Note that CFC epitomizes prevalent techniques used for recommendation fairness, strictly applying an adversarial constraint at the embedding level to obscure sensitive attributes. This strategy aims to deceive classifiers reliant on these learned representations. In this sense, CFC's prediction bias can be viewed as a lower bound for fairness metric, regardless of its recommendation performance. From the results, we find:

- **From the perspective of performance, AdvDrop exhibits a substantial performance boost over the LightGCN backbone and sustains a prediction bias comparable to MF.** This contrasts notably with many conventional fairness-centric methods which often sacrifice model efficacy to minimize prediction bias. Moreover, introducing attribute information at the embedding layer (+Embed info) or further constraining $P_B$ based on attribute categories (+Mask Info) allows AdvDrop to further enhance the recommendation performance while reducing prediction bias, approaching the CFC's fairness metric lower bound.
- **From the perspective of representations, AdvDrop alleviates undesired clustering of user/items sharing sensitive attributes and moderates the emphasis on trending items during neighbor aggregation.** Specifically, we can view the item popularity and user gender as the attributes of items and users, and plot T-SNE visualization of learned user/item representations w.r.t each attribute in Figure 1. For the user gender, clear clustering trends manifest from Figures 1a, 1b to 1c 1d, indicating that the bias *w.r.t.* gender amplifies with the increase of graph convolution layers. In contrast, representations from AdvDrop exhibit a more dispersed pattern, akin to MF, showing that AdvDrop shields the amplification from the GNN mechanism.

These two observations together clearly demonstrate that AdvDrop mitigates the intrinsic bias in graph-based CF. This not only enhances the model generalization, but also encourages the fairness of recommendation.

## 4.4 Study of AdvDrop (RQ3)

*4.4.1 Ablation Study.* To further investigate the effectiveness of AdvDrop's components, we conduct an ablation study on Yelp2018 and present the results in Table 4. We observe that:

- **When removing the adversarial learning of $P_B$** (denoted as "w/o $P_B$"), the ID performance slightly drops (with NDCG@20 decreasing from 0.817 to 0.781), while the OOD performance sees a more signification reduction (with NDCG@20 decreasing from 0.0073 to 0.0060). Note that such an ablated model, leveraging only the contrastive loss $\mathcal{L}_{inv}$ from two randomly dropout graph views, is similar to the recent models like RDrop [31] and SGL [56]. The observed performance boost compared to the graph-based backbone can be attributed to the data augmentations achieved through crafting random views of the original interaction graph.
- **When further discarding the contrastive learning objective $\mathcal{L}_{inv}$** (denoted as "w/o $P_B$ & $\mathcal{L}_{inv}$"), the model is restricted to learn solely via random graph dropout, leading to performance metrics falling short of those achieved with the graph-based backbone.

This evidence implies that **the superior performance of AdvDrop results from both learning stages**: the Debias Representation Learning stage fundamentally improves representation quality, while the Bias Identification stage actively seeks meaningful bias-related views that further boost model generalization performance.

*4.4.2 Visualization of Bias Measurement Function $P_B$.* To understand the crucial information captured by the Bias Identification stage, we visualize the learned bias measurement function $P_B$ *w.r.t.* item popularity. Specifically, we sort all items based on popularity, then split them into four groups from 0 to 3 with ascending popularity. We compute the average $P_B$ of interactions connected to items within each group, shown in in Figure 4. Clearly, only the

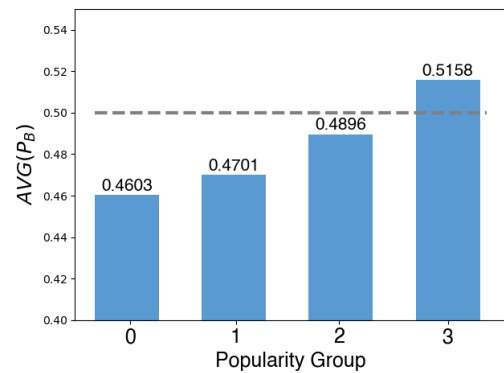

Figure 4: Visualization of learned bias measurement function $P_B$ *w.r.t.* item popularity.

interactions associated with the top quartile (group 3) — representing the most popular items — have an average $P_B$ exceeding 0.5. In contrast, the other three quartiles have a $P_B$ average below this threshold. Moreover, there's a clear trend: the average $P_B$ increases congruent with item popularity. These insights suggest that the model not only learns $P_B$ with a prudent tendency around 0.5 but also adeptly mirrors the long-tailed distribution of user-item interactions. This indicates the Bias Identification stage effectively captures the information about interaction bias.

*4.4.3 Interpretation of AdvDrop for Debiased Learning.* We present an intriguing interpretation of how AdvDrop effectively conducts general debiasing to achieve superior performance. Figure 5 shows The relationship between recommendation performance (measured by NDCG@3) and model bias (quantified by prediction bias) during training on Coat for MF, LightGCN, and AdvDrop. Our analysis yielded the following insights:

- Both MF and LightGCN's recommendation performance improves consistently with more training steps. This suggests that as they enhance recommendation performance during training, they also accrue representational bias. Notably, the LightGCN demonstrated superior NDCG@3 compared to MF, albeit with a higher degree of prediction bias, reflecting its heightened recommendation accuracy and exacerbated inherent bias.
- Contrary to a consistent improvement, AdvDrop's performance experienced a slight decrease during training before recovering and subsequently improving, whereas the prediction bias first accumulated but was subsequently mitigated. This trajectory indicates AdvDrop's ability to iteratively optimize by first reducing the bias and then enhancing the recommendation performance. Tracing AdvDrop's training might reveal a sequence: an initial boost in recommendation performance with bias accumulation, a subsequent performance dip with continued bias accumulation or at bias removal's onset, and a final surge in performance once the bias is eradicated.
- For AdvDrop, the time order between re-improvement of recommendation performances and bias removal differs across different attributes. Specifically, the bias mitigation either precedes, coincides with, or follows performance enhancements for various attributes, respectively. This indicates that the bias identification

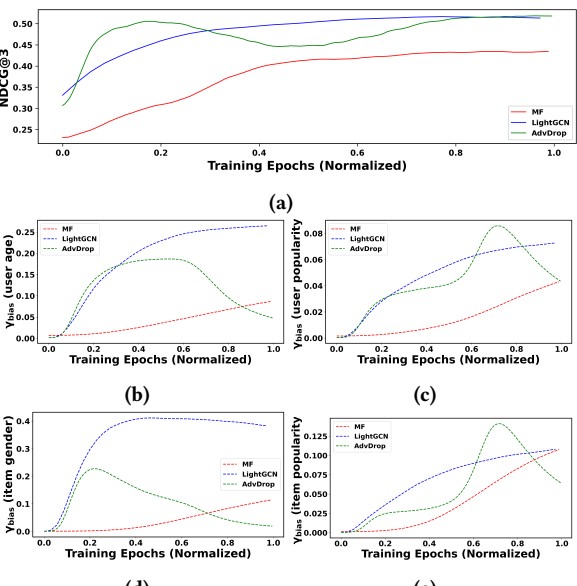

Figure 5: (a) The overall recommendation performance v.s. epochs during training. (b) ~ (e) The debiasing performance *w.r.t.* epochs on user age, user popularity, item gender, and item popularity attributes, respectively.

and removal in AdvDrop is conducted in an ordered manner, and the superior performance can be attributed to the ensemble of orderly removal for different biases.

In a nutshell, these observations clearly depict the bias identification and bias removal process in AdvDrop, accounting for its superiority. The orderly removal of bias in AdvDrop is an interesting phenomenon that possibly implies implicit ordering in different recommendation biases, which we will explore in future work.

## 5 CONCLUSION

In this work, we proposed a novel framework AdvDrop, which is designed to alleviate both general biases and inherent bias amplification in graph-based CF, by enforcing embedding-level invariance from learned bias-related views. Grounded by extensive experiments and interpretable visualization, AdvDrop successfully identifies various bias factors and performs iterative bias removal to achieve superior OOD generalization performance for recommendation. For future work, it would be worthwhile to design algorithms that consider bias-related views for both data and graph, or to explore the theoretical guarantee for convergence of AdvDrop. We believe that AdvDrop points in a promising direction for general debiasing in graph-based CF models and will inspire more work in the future.

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

# A  APPENDIX

## A.1  Related Work

### A.1.1  *Specific Debiasing in recommender systems*. . Recommender systems usually face various bias issues due to the discrepancies between observed behavioral data and users' true preferences. The common biases include 1) Popularity Bias 2) Exposure Bias, 3) Conformity Bias, and 4) Unfairness.

**Popularity Bias.** Recommender systems often exhibit a bias towards popular items, as these items are frequently presented to users and thus have a higher likelihood of being clicked. Various methods have been proposed to counteract this popularity bias: 1) Regularization-based frameworks [1, 16, 17, 55] balance the trade-off between accuracy and coverage by incorporating penalty terms. For instance, ESAM [17] leverages center-wise clustering and self-training regularization to enhance the influence of long-tail items. CDAN [16] employs the Pearson coefficient correlation as a regularization measure to separate item property representations from their popularity. 2) Sample re-weighting methods [8, 24, 27, 41] adjust the loss of each instance by inversely weighting the item propensity score in the training dataset, also known as IPS. Given that propensity scores in IPS approaches can exhibit high variance, many studies [8, 24] have turned to normalization or smoothing penalties to ensure model stability. Recent works [45, 54, 55, 57] have drawn inspiration from Stable Learning and Causal Inference. For instance, MACR [54] conducts counterfactual inference using a causal graph, postulating that popularity bias originates from the item node influencing the ranking score. Meanwhile, sDRO [55] integrates a Distributionally Robust Optimization (DRO) framework to minimize loss variances in long-tailed data distributions. However, devising these causal graphs and understanding the environmental context often hinge on heuristic insights from researchers.

**Exposure Bias.** User behaviors are easily affected by the exposure policy of a recommender system, which deviates user actions from true preference. To address exposure bias, researchers have proposed two primary methods: 1) Reweighting methods [30, 40, 62] treat all the unobserved interactions as negative and reweight them by specifying their confidence scores. For instance, AMAN [30] defines the user-item feature similarity as the confidence score. UBPR [40] proposes a new weighting strategy with a propensity score to estimate confidence. 2) Causal Inference methods [46, 57, 58] mitigate exposure bias by leveraging counterfactual inference. For example, DCCF [57] utilizes a specific causal technique, forward door criterion, to mitigate the effects of unobserved confounders.

**Conformity Bias.** Conformity bias arises as users within a group display similar behaviors, even if such actions deviate from their genuine preferences. To counteract this bias, leading debiasing methods roughly fall into two main categories: 1) Modeling popularity influence methods [32, 66, 67] aim to counteract conformity bias by factoring them in popularity. CM-C [32] utilizes previous ratings to estimate and predict unknown ratings, considering group size, cohesion, and unanimity factors. DICE [67] disentangle conformity embeddings and interest embeddings in the popularity perspective to enforce the model invariant to conformity bias. 2) Modeling social influence methods [10, 33, 49] consider the user's

ratings as user preference and social influence. In particular, PTPMF [49] proposes a Probabilistic Matrix Factorization model that considers the distinction between strong and weak social ties to learn personalized preference.

**Unfairness.** Unfairness in recommender systems is often attributed to the system's predisposition to generate biased predictions and representations concerning specific user or item attributes. Efforts to tackle this unfairness have led to several methodologies, primarily falling under three categories: 1) Rebalancing-based methods [20, 22, 37, 43] draw inspiration from solutions to the class imbalance problem, focusing on balancing recommendation outputs in relation to sensitive attributes. For instance, HyPER [20] constructs user-user and item-item similarity measures while considering content and demographic data to balance discrepancies across groups. Fairwalk [37] utilizes random walks on graph structures, leveraging sensitive attributes to derive unbiased embeddings. 2) Regularization-based frameworks [28, 59, 60] integrate fairness criteria as regularizers, aiming to minimize group discrepancies. An exemplary model, IERS [28] devises a fairness regularizer that factors in the expected independence between sensitive attributes and the resultant predictions. 3) Adversarial learning-based frameworks[4, 6, 70] operate by alternately optimizing a primary prediction model and an adversarial model dedicated to debiasing. For instance, CFC [6] employs filters to extract sensitive information and counteracts these with discriminators, which attempt to predict sensitive attributes from the sanitized embeddings.

*A.1.2 General Debiasing in recommender systems.* It aims to address multiple underlying data biases in a dataset simultaneously. Without knowing the exact type of bias, general debiasing requires the model to yield satisfactory performance on various bias-related distributions. However, general debiasing in recommender systems remains largely unexplored. In the early stage, DR [51] utilizes a small part of missing-at-random data in the test set to generally mitigate various biases in the data set. CausE [5] introduces a domain adaptation algorithm to extract insights from logged data that has been subjected to random exposure. More recent research trends focus on the active identification of latent bias structures followed by their removal [52, 53]. For instance, InvPref [53] proposes to learn invariant embeddings and perform environment label assignment alternately. Motivated by the recent success of invariant learning in computer vision and natural language processing areas, we believe our work is the first to enforce embedding-level invariance *w.r.t.* adversarially learned views of interaction graphs for general debiasing.

## A.2 Datasets

- **Yahoo [35] & Coat [41].** Both datasets are commonly used as benchmarks for general debiasing in recommendation. They both consist of a biased training set of normal user interactions and an unbiased uniform test set collected by a random logging policy. During data collection, users interact with items by giving ratings (1-5). In our experiments, interactions with scores $\geq 4$ are considered positive samples, and negative samples are collected from all possible user-item pairs.

- **KuaiRec [21].** This dataset originates from real-world recommendation logs of KuaiShou, a prominent short video-sharing platform. Distinctively, the testing dataset contains dense ratings, encompassing feedback from a total of 1,411 users across 3,327 items, whereas the training dataset is relatively sparse. We treat items with a viewing duration that surpasses twice the stipulated length of the corresponding short video as constituting positive interactions.

- **Yelp2018 [26].** This dataset is adopted from the 2018 edition of the Yelp challenge. Users visiting local businesses are recorded as interactions. Following previous work[63], we split the dataset into in-distribution (ID) and out-of-distribution (OOD) test splits with respect to popularity.

## A.3 Evaluation Metrics

- **NDCG@K** measures the ranking quality by discounting importance based on position and is defined as below:

$$DCG_u@K = \sum_{(u,v) \in D_{test}} \frac{I(\hat{r}_{u,i} \leq K)}{\log(\hat{r}_{u,v} + 1)}$$

$$NDCG_u@K = \frac{1}{|\mathcal{U}|} \sum_{u \in \mathcal{U}} \frac{DCG_u@K}{IDCG_u@K},$$

where $D_{test}$ is the test dataset, $\hat{r}_{u,i}$ is the rank of item i in the list of relevant items of u, $\mathcal{U}$ is the set of all users, and $IDCG_u@K$ is the ideal $DCG_u@K$.

- **Recall@K** measures the percentage of the recommended items in the user-interacted items. The formula is as below:

$$Recall_u@K = \frac{\sum_{(u,i) \in D_{test}} I(\hat{r}_{u,i} \leq K)}{|D^u_{test}|}$$

$$Recall@K = \frac{1}{|\mathcal{U}|} \sum_{u \in \mathcal{U}} Recall_u@K,$$

where $D^u_{test}$ is the set of all interactions of user u in the test dataset $D_{test}$.

- **Prediction bias** measure the level of parity regarding predicted recommendation ratings w.r.t. a given attribute. Prediction bias of a certain user attribution is given by:

$$\gamma_{bias} = \frac{1}{|\mathcal{I}|} \sum_{i \in \mathcal{I}} \max_{\substack{a_1, a_2 \in A \\ a_1 \neq a_2}} \left| \text{Avg}_{\{u|a_u=a_1\}}(\hat{y}_{ui}) - \text{Avg}_{\{v|a_v=a_2\}}(\hat{y}_{vi}) \right|,$$

where $a_1, a_2$ are two user attribute labels, $A$ is the set of user attribute labels, and $\mathcal{I}$ is the set of items. Item attribute's prediction bias can be obtained likewise.

## A.4 Baselines

We compare with general debiasing strategies in various research lines. We also compare with representative methods for popularity bias and fairness issues. The adopted baselines include:

- **IPS-CN [24]**: IPS [36] re-weights training samples inversely to the estimated propensity score. IPS-CN adds clipping and normalization on plain IPS to achieve lower variance. For a fair comparison, we implemented IPS-CN with propensity score according to item popularity in the training set without introducing other user/item attributes.

**Table 5: Comparison of general debiasing performance on Coat, Yahoo, and KuaiRec. The improvements over the baselines are statistically significant at 0.05 level ($p$-value $\ll 0.05$).**

| | Coat | | Yahoo | | KuaiRec | |
|---|---|---|---|---|---|---|
| | NDCG@5 | Recall@5 | NDCG@5 | Recall@5 | NDCG@30 | Recall@30 |
| LightGCN | 0.523 | 0.519 | 0.673 | 0.804 | 0.331 | 0.114 |
| IPS-CN | $\underline{0.538}^{+2.87\%}$ | $0.524^{+0.96\%}$ | $0.664^{-1.34\%}$ | $0.797^{-0.87\%}$ | $0.014^{-95.77\%}$ | $0.003^{-97.37\%}$ |
| DR | $0.524^{+0.19\%}$ | $\underline{0.534}^{+2.89\%}$ | $\underline{0.677}^{+0.59\%}$ | $\underline{0.805}^{+0.12\%}$ | $0.035^{-89.43\%}$ | $0.013^{-88.60\%}$ |
| CVIB | $0.528^{+0.96\%}$ | $\underline{0.538}^{+3.66\%}$ | $0.657^{-2.38\%}$ | $0.787^{-2.11\%}$ | $\underline{0.336}^{+1.51\%}$ | $\underline{0.118}^{+3.51\%}$ |
| InvPref | $0.421^{-19.50\%}$ | $0.438^{-15.61\%}$ | $0.653^{-2.97\%}$ | $0.773^{-3.86\%}$ | - | - |
| AutoDebias | $0.529^{+1.15\%}$ | $0.527^{+1.54\%}$ | $0.658^{-2.23\%}$ | $0.782^{-2.74\%}$ | $0.316^{-4.53\%}$ | $0.105^{-7.89\%}$ |
| **AdvDrop** | $\mathbf{0.553}*^{+5.74\%}$ | $\mathbf{0.540}*^{+4.05\%}$ | $\mathbf{0.681}*^{+1.19\%}$ | $\mathbf{0.807}*^{+0.37\%}$ | $\mathbf{0.351}*^{+6.04\%}$ | $\mathbf{0.130}*^{+14.04\%}$ |

- **DR** [51]: This method combines a data imputation method that assigns predefined scores to interactions with basic IPS. Unbiased learning can be achieved as long as one of the components is accurate.
- **CVIB** [52]: This method incorporates a contrastive information loss and an additional output confidence penalty, which facilitates balanced learning between factual and counterfactual domains to achieve unbiased learning.
- **InvPref** [53]: It iteratively decomposed the invariant preference and variant preference by estimating heterogeneous environments adversarially, which is the first attempt to actively identify and remove latent bias for general debia purposes.
- **AutoDebias** [11]: This model leverages a small set of uniform data to optimize the debiasing parameters with meta-learning, followed by utilizing the parameters to guide the learning of the recommendation model.
- **CDAN** [16]: This model uses Pearson coefficient correlation as regularization to disentangle item property representations from item popularity representation, and introduces additional unexposed items to align prediction distributions for head and tail items.
- **sDRO** [55]: This model adds streaming optimization improvement to the Distributionally Robust Optimization (DRO) framework, which mitigates the amplification of Empirical Risk Minimization on popularity bias.
- **CFC** [6]: This model tackles representation level unfairness in recommendation by adversarially training filters for removing sensitive information against discriminators that predict sensitive attributes from filtered embeddings.

## A.5 Implementation Details

In our experiments, we trained all models on a single Tesla-V100 GPU with the number of layers of LightGCN set to 2. *Adam* is used as the optimization algorithm for both learning stages. For the Debiased Representation Learning stage, we randomly sampled 100 users/items as negative samples and set $\tau = 0.1$ for the contrastive objective. The coefficient $\lambda$ that combines the recommendation objective and contrastive objective is set to 1. Other hyperparameters regarding training batch size, embedding size, and epochs & learning rate for both stages on different datasets are shown in table 7. Batch size and embedding size for training are fixed on each individual dataset across all compared methods.

**Table 6: Dataset statistics.**

| | Coat | Yahoo | KuaiRec(train) | KuaiRec(test) | Yelp2018 |
|---|---|---|---|---|---|
| #Users | 290 | 14,382 | 7176 | 1411 | 4886 |
| #Items | 295 | 1000 | 10,729 | 3327 | 4804 |
| #Interactions | 2776 | 5,397,926 | 12,530,806 | 4,676,570 | 134,031 |
| Density | 0.032 | 0.009 | 0.134 | 0.996 | 0.006 |

**Table 7: Hyper-parameters of AdvDrop on different datasets.**

| | AdvDrop hyper-parameters | | | | | |
|---|---|---|---|---|---|---|
| | $K_{stage1}$ | $K_{stage2}$ | $lr_{main}$ | $lr_{adv}$ | embed_size | batch_size |
| Coat | 7 | 10 | 1e-3 | 1e-2 | 30 | 128 |
| Yahoo | 15 | 5 | 31e-3 | 1e-3 | 30 | 128 |
| KuaiRec | 3 | 5 | 5e-4 | 1e-3 | 30 | 512 |
| Yelp2018 | 7 | 15 | 5e-4 | 1e-2 | 64 | 1024 |

