# OpenReview forum: "General Debiasing for Graph-based Collaborative Filtering via Adversarial Graph Dropout"
_ACM.org/TheWebConf/2024/Conference — TheWebConf24_

### Official Review · Reviewer_v2Fm · 2023-11-20

**Novelty:** 5
**Technical Quality:** 5

**Review:**

This paper proposed a novel debiasing framework named Adversarial Graph Dropout (AdvDrop), aimed at addressing inherent biases in graph-based Collaborative Filtering (CF) models. The authors initially identify and highlight the issue of biases in graph-based CF models, particularly the biases amplified during the aggregation of user-item interaction information using Graph Neural Networks (GNNs). They then propose the AdvDrop framework, which employs adversarial learning to distinguish between biased and unbiased interactions and ensures that the representations aggregated from these interactions remain invariant. Through a series of extensive experiments, including tests across different bias scenarios and visualization of representations, the authors validate the effectiveness of AdvDrop in mitigating inherent biases, enhancing model generalization, and promoting fairness in recommendations.

Strengths:

1. The study addresses a significant and non-negligible problem in recommender systems - inherent biases. This is crucial for enhancing the fairness and accuracy of recommendation systems, directly impacting the user experience and satisfaction.

2. The paper not only proposes a new solution but also delves deeply into the root causes of the problem and the different types of biases. This thorough exploration of the issue contributes to further research in the field.

3. The paper is well-written, with a clear logical structure and expression, making it easy for readers to understand complex concepts and methodologies.

Weaknesses:

1. While the paper proposes a novel solution, the discussion on the theoretical underpinnings and in-depth mathematical principles behind this solution might not be sufficiently thorough. Adding more theoretical analysis could help in better understanding how the framework operates and its limitations.

2. The paper might not delve deeply into the long-term impacts of the AdvDrop framework on the performance of recommendation systems and user behavior. Understanding these long-term effects is crucial for assessing the sustainability of the framework in practical applications.

3. While the focus of the research is on reducing algorithmic bias, it may not fully consider the impact of this debiasing approach on the end-user experience. User satisfaction and engagement are key factors in measuring the success of recommendation systems and should be considered in future research.

**Questions:**

1. The theoretical basis of the AdvDrop framework seems to be insufficiently discussed in the paper. Could you elaborate on the mathematical principles and assumptions behind this framework? Specifically, how is the role and mechanism of adversarial learning in debiasing concretely implemented?

2. There seems to be a lack of assessment in the paper on the long-term impacts of the AdvDrop framework. Have you considered the potential effects this framework might have on the long-term performance of recommendation systems and user behavior?

3. How do you balance algorithmic fairness with personalized user experience in the process of debiasing? Does the AdvDrop framework negatively impact user satisfaction and engagement while reducing biases?

**Reviewer Confidence:**

3: The reviewer is confident but not certain that the evaluation is correct

**Scope:**

4: The work is relevant to the Web and to the track, and is of broad interest to the community

---

### Official Review · Reviewer_zGW2 · 2023-11-21

**Novelty:** 4
**Technical Quality:** 4

**Review:**

This paper proposes an approach to optimize subgraph sampling through adversarial learning, thereby partitioning the original user-item interaction graph into two graphs: one that is bias-aware and another that mitigates bias. This enables the learning of debiased representations for recommendation purposes.

Strength:
1. This article addresses a crucial issue: how to mitigate biases present in recommendation.
2. The experiment seems to show the effectiveness of proposed method.

Weakness:

Some questions regarding the methods and motivation in this paper need to be addressed. Please refer to the "Questions" section for further clarification.

**Questions:**

Q1: In the visual results presented in Figure 1, the effectiveness of AdaDrop is not clearly demonstrated. On the one hand, in Figure 1(a)-1(d), the features of Male and Female remain clearly separable; on the other hand, in Figure 1(e)-2(g), embeddings for the Tail category are also clustered together.

Q2: The author mentions the presence of various bias factors in real-world situations. A question that arises is whether the simple partitioning into only two sub-graphs (i.e., bias-aware and bias-mitigated) is feasible considering there are many bias factors. Perhaps in cases where only one bias factor is considered, this approach is feasible because It allows for a clear separation of whether the edge is influenced by that specific bias factor. However, the author considers multiple bias factors, raising the question of whether these factors may influence and entangle with each other. In such a scenario, can we still be sure that partitioning into only two sub-graphs for all the bias factors is enough?

Q3: Is minimizing $\mathcal{L}_{inv}$ necessary when optimizing the recommenders? Since the method is going to encode two different representations—one for bias-aware and another for bias-mitigated—why should we use contrastive learning to bring these two different representations close for the same node in the graph? What exactly is the 'invariant' part for, given that the two graphs are different? Why is there an invariant part?

Q4: Is there something like a case study to further convince that the bias is alleviated?

**Reviewer Confidence:**

3: The reviewer is confident but not certain that the evaluation is correct

**Scope:**

4: The work is relevant to the Web and to the track, and is of broad interest to the community

---

### Official Review · Reviewer_QSnU · 2023-11-21

**Novelty:** 6
**Technical Quality:** 6

**Review:**

The paper presents a novel embedding framework for recommender systems that identifies bias present in user-item interactions and also removes this bias by randomly sampling subgraphs. The authors discuss their approach theoretically and then illustrate and evaluate their approach on a collection of datasets. The experimental results show increased performance as compared to a set of baseline and sota methods. Finally, the authors analyze their approach by performing an ablation study and by analyzing the representations obtained by their approach. The paper is generally very well written and has promising results. Potentially, the paper may be impactful on future research and development of debiasing methods in recommender systems.

**Questions:**

1) Figure 4 show the probabilities of dropping interactions for various popularity groups. Essentially, this probability is proportional to popularity. I would expect similar probabilities for dropping interactions w.r.t. sensitive attributes. Hence, we can think of a simple baseline for removing popularity or minority/majority biases: we just simply drop interactions with probability proportional to a bias that we want to remove (without any training, just using relative frequencies). What is the improvement of the presented approach over such a baseline?

**Reviewer Confidence:**

3: The reviewer is confident but not certain that the evaluation is correct

**Scope:**

4: The work is relevant to the Web and to the track, and is of broad interest to the community

---

### Official Review · Reviewer_X8fj · 2023-11-22

**Novelty:** 4
**Technical Quality:** 4

**Review:**

# summary
This paper proposes AdvDrop, a framework that addresses biased representation learning in graph neural networks (GNNs) used for collaborative filtering (CF) in recommender systems. AdvDrop differentiates between unbiased and biased interactions and employs adversarial learning to split the neighborhood into bias-mitigated and bias-augmented views. By aggregating information separately and ensuring representation invariance, AdvDrop mitigates bias and achieves unbiased representations. Experimental results on various datasets demonstrate its effectiveness in improving CF models and separating subgraphs.

# strengths
- This paper is well-organized and provides open-source code.
- Using adversarial methods to mitigate bias issues in GNNs is a very promising approach.

# weaknesses
- The novelty of the proposed method is limited, and some concepts and definitions are confusing.
- The comparative experiments yielded unsatisfactory results and lacked convincing evidence.

**Questions:**

1、While the theoretical aspects of this paper are well-described, I recommend further clarification on the specific details related to adversarial learning. This includes providing a more explicit explanation of the training hyperparameters and their significant impact on the training performance. Enhancing the clarity of these details is essential for readers to better understand and follow, ultimately improving the overall effectiveness of the proposed approach.

2、The experimental results were conducted on three public datasets, but the performance of both the baseline and the proposed method on the largest dataset, kuairec, was not particularly satisfactory. Would it be necessary to conduct experiments on more suitable datasets? The experimental results fail to effectively persuade me and demonstrate the superiority of this method.

**Ethics Review Description:**

\

**Reviewer Confidence:**

3: The reviewer is confident but not certain that the evaluation is correct

**Scope:**

3: The work is somewhat relevant to the Web and to the track, and is of narrow interest to a sub-community

---

### Official Review · Reviewer_2uHt · 2023-11-30

**Novelty:** 3
**Technical Quality:** 3

**Review:**

The paper discusses the issue of biased representation learning in graph-based collaborative filtering (CF) models used in recommender systems. It highlights that the aggregation mechanism in these models amplifies biases present in the user-item interaction graph, leading to distorted views of users and items. To address this problem, the authors propose a novel framework called Adversarial Graph Dropout (AdvDrop), which differentiates between biased and unbiased interactions and enables unbiased representation learning.

AdvDrop employs adversarial learning to split the neighborhood into two views: one with bias-mitigated interactions and the other with bias-aware interactions. After view-specific aggregation, the framework ensures that the bias-mitigated and bias-aware representations remain invariant, mitigating the influence of bias. Experimental results on various datasets demonstrate the effectiveness of AdvDrop in reducing biases and improving recommendation accuracy.

A potential negative point of this work is the lack of detailed descriptions of parameter tuning for the compared baseline methods. While the paper compares AdvDrop with existing debiasing baselines, it does not provide comprehensive information on how the parameters of these baselines were tuned. Without a clear description of the parameter tuning process, it becomes difficult to assess the fairness of the comparison and understand whether the baselines were optimized for their best performance.

The paper lacks a comprehensive analysis of the computational efficiency of the proposed AdvDrop framework. Considering the potential scalability challenges of graph-based models, it would have been valuable to investigate and report the computational overhead introduced by AdvDrop, particularly in terms of training time and memory requirements. Such analysis would help readers assess the practical feasibility and scalability of AdvDrop in real-world scenarios with large-scale graphs and datasets.

**Questions:**

This work proposes the Adversarial Graph Dropout (AdvDrop) framework to address the issue of biased representation learning in graph-based collaborative filtering models. The paper highlights the amplification of biases in the aggregation mechanism of these models and introduces AdvDrop as a solution. AdvDrop differentiates between biased and unbiased interactions and employs adversarial learning to split the neighborhood into bias-mitigated and bias-aware views. The paper demonstrates the effectiveness of AdvDrop through experiments on public datasets, showcasing reduced biases and improved recommendation accuracy. However, the work lacks in-depth analysis of the proposed framework, comparison with state-of-the-art methods, evaluation on real-world datasets, and detailed descriptions of parameter tuning for baseline methods. Additionally, the computational efficiency and potential unintended consequences of AdvDrop are not thoroughly discussed.

**Reviewer Confidence:**

3: The reviewer is confident but not certain that the evaluation is correct

**Scope:**

4: The work is relevant to the Web and to the track, and is of broad interest to the community

---

### Decision · Program_Chairs · 2024-01-22

**Decision:**

Accept

**Comment:**

Summary: this paper introduces AdvDrop, a novel framework designed to mitigate bias in graph neural networks (GNNs) used for collaborative filtering in recommender systems. It employs adversarial learning to separate user-item interactions into bias-mitigated and bias-augmented views, leading to unbiased representation learning. The effectiveness of AdvDrop is demonstrated through extensive experiments across various datasets.

 #### Strengths
 1. **Innovative Approach**: Utilizes adversarial methods to effectively address bias issues in GNNs.
 2. **Comprehensive Structure**: Well-organized presentation and provision of open-source code for replication.

 #### Weaknesses
 1. **Limited Novelty**: Some concepts are not novel and are confusingly defined.
 2. **Unconvincing Experimental Results**: The comparative experiments lack compelling evidence of the method's superiority.
 3. **Lack of Theoretical Depth**: Insufficient discussion on the mathematical principles and long-term impacts of the AdvDrop framework.
 4. **User Experience Considerations**: The impact of debiasing on user satisfaction and engagement is not fully addressed.

 #### Suggestions for Improvement
 1. **Clarify Adversarial Learning Details**: Provide explicit explanations of training hyperparameters and their impact on performance.
 2. **Dataset Suitability**: Consider experimenting with more suitable datasets, especially where current results are not satisfactory.
 3. **Enhance Theoretical Discussion**: Expand on the theoretical basis of AdvDrop, including its mathematical principles and assumptions.
 4. **Long-Term Impact Analysis**: Assess the potential long-term effects of AdvDrop on recommendation system performance and user behavior.
 5. **User Experience Balance**: Investigate how AdvDrop balances algorithmic fairness with personalized user experience, and its effect on user satisfaction.
 6. **Compare with Simple Baselines**: Justify the improvement of AdvDrop over simpler baselines like dropping interactions proportional to bias.
 7. **Improve Clarity and Consistency**: Address issues with clarity in figures and consistency in explanations and representations.

 The authors made good efforts in responding and summarizing reviewer's questions, which ultimately weighs positively at ACs side and impacts my final recommendation.